# Structured Generative Adversarial Networks

[1]**Zhijie Deng**[*], [2,3]**Hao Zhang**[*], [2]**Xiaodan Liang**, [2]**Luona Yang,**
[1,2]**Shizhen Xu**, [1]**Jun Zhu**[†], [3]**Eric P. Xing**
[1]Tsinghua University, [2]Carnegie Mellon University, [3]Petuum Inc.
{dzj17,xsz12}@mails.tsinghua.edu.cn, {hao,xiaodan1,luonay1}@cs.cmu.edu,
dcszj@mail.tsinghua.edu.cn, epxing@cs.cmu.edu

## Abstract

We study the problem of conditional generative modeling based on designated semantics or structures. Existing models that build conditional generators either require massive labeled instances as supervision or are unable to accurately control the semantics of generated samples. We propose structured generative adversarial networks (SGANs) for semi-supervised conditional generative modeling. SGAN assumes the data $x$ is generated conditioned on two independent latent variables: $y$ that encodes the designated semantics, and $z$ that contains other factors of variation. To ensure disentangled semantics in $y$ and $z$, SGAN builds two collaborative games in the hidden space to minimize the reconstruction error of $y$ and $z$, respectively. Training SGAN also involves solving two adversarial games that have their equilibrium concentrating at the true joint data distributions $p(x, z)$ and $p(x, y)$, avoiding distributing the probability mass diffusely over data space that MLE-based methods may suffer. We assess SGAN by evaluating its trained networks, and its performance on downstream tasks. We show that SGAN delivers a highly controllable generator, and disentangled representations; it also establishes start-of-the-art results across multiple datasets when applied for semi-supervised image classification (1.27%, 5.73%, 17.26% error rates on MNIST, SVHN and CIFAR-10 using 50, 1000 and 4000 labels, respectively). Benefiting from the separate modeling of $y$ and $z$, SGAN can generate images with high visual quality and strictly following the designated semantic, and can be extended to a wide spectrum of applications, such as style transfer.

## 1 Introduction

Deep generative models (DGMs) [12, 8, 26] have gained considerable research interest recently because of their high capacity of modeling complex data distributions and ease of training or inference. Among various DGMs, variational autoencoders (VAEs) and generative adversarial networks (GANs) can be trained unsupervisedly to map a random noise $z \sim \mathcal{N}(\mathbf{0}, \mathbf{1})$ to the data distribution $p(x)$, and have reported remarkable successes in many domains including image/text generation [17, 9, 3, 27], representation learning [27, 4], and posterior inference [12, 5]. They have also been extended to model the conditional distribution $p(x|y)$, which involves training a neural network generator $G$ that takes as inputs both the random noise $z$ and a condition $y$, and generates samples that have desired properties specified by $y$. Obtaining such a conditional generator would be quite helpful for a wide spectrum of downstream applications, such as classification, where synthetic data from $G$ can be used to augment the training set. However, training conditional generator is inherently difficult, because it requires not only a holistic characterization of the data distribution, but also fine-grained alignments between different modes of the distribution and different conditions. Previous works have tackled this problem by using a large amount of labeled data to guide the generator's learning [32, 23, 25], which compromises the generator's usefulness because obtaining the label information might be expensive.

[*] indicates equal contributions. [†] indicates the corresponding author. 31st Conference on Neural Information Processing Systems (NIPS 2017), Long Beach, CA, USA.

In this paper, we investigate the problem of building conditional generative models under semi-supervised settings, where we have access to only a small set of labeled data. The existing works [11, 15] have explored this direction based on DGMs, but the resulted conditional generators exhibit inadequate *controllability*, which we define as the generator's ability to conditionally generate samples that have structures strictly agreeing with those specified by the condition – a more controllable generator can better capture and respect the semantics of the condition.

When supervision from labeled data is scarce, the controllability of a generative model is usually influenced by its ability to disentangle the designated semantics from other factors of variations (which we will term as *disentanglability* in the following text). In other words, the model has to first learn from a small set of labeled data what semantics or structures the condition $y$ is essentially representing by trying to recognize $y$ in the latent space. As a second step, when performing conditional generation, the semantics shall be *exclusively* captured and governed within $y$ but not interweaved with other factors. Following this intuition, we build the structured generative adversarial network (SGAN) with enhanced controllability and disentanglability for semi-supervised generative modeling. SGAN separates the hidden space to two parts $y$ and $z$, and learns a more structured generator distribution $p(x|y, z)$ – where the data are generated conditioned on two latent variables: $y$, which encodes the designated semantics, and $z$ that contains other factors of variation. To impose the aforementioned exclusiveness constraint, SGAN first introduces two dedicated inference networks $C$ and $I$ to map $x$ back to the hidden space as $C : x \rightarrow y, I : x \rightarrow z$, respectively. Then, SGAN enforces $G$ to generate samples that when being mapped back to hidden space using $C$ (or $I$), the inferred latent code and the generator condition are always matched, regardless of the variations of the other variable $z$ (or $y$). To train SGAN, we draw inspirations from the recently proposed adversarially learned inference framework (ALI) [5], and build two adversarial games to drive $I, G$ to match the true joint distributions $p(x, z)$, and $C, G$ to match the true joint distribution $p(x, y)$. Thus, SGAN can be seen as a combination of two adversarial games and two collaborative games, where $I, G$ combat each other to match joint distributions in the visible space, but $I, C, G$ collaborate with each other to minimize a reconstruction error in the hidden space. We theoretically show that SGAN will converge to desired equilibrium if trained properly.

To empirically evaluate SGAN, we first define a *mutual predictability* (MP) measure to evaluate the disentanglability of various DGMs, and show that in terms of MP, SGAN outperforms all existing models that are able to infer the latent code $z$ across multiple image datasets. When classifying the generated images using a golden classifier, SGAN achieves the highest accuracy, confirming its improved controllability for conditional generation under semi-supervised settings. In the semi-supervised image classification task, SGAN outperforms strong baselines, and establishes new state-of-the-art results on MNIST, SVHN and CIFAR-10 dataset. For controllable generation, SGAN can generate images with high visual quality in terms of both visual comparison and inception score, thanks to the disentangled latent space modeling. As SGAN is able to infer the unstructured code $z$, we further apply SGAN for style transfer, and obtain impressive results.

## 2 Related Work

DGMs have drawn increasing interest from the community, and have been developed mainly toward two directions: VAE-based models [12, 11, 32] that learn the data distribution via maximum likelihood estimation (MLE), and GAN-based methods [19, 27, 21] that train a generator via adversarial learning. SGAN combines the best of MLE-based methods and GAN-based methods which we will discuss in detail in the next section. DGMs have also been applied for conditional generation, such as CGAN [19], CVAE [11]. DisVAE [32] is a successful extension of CVAE that generates images conditioned on text attributes. In parallel, CGAN has been developed to generate images conditioned on text [24, 23], bounding boxes, key points [25], locations [24], other images [10, 6, 31], or generate text conditioned on images [17]. All these models are trained using fully labeled data.

A variety of techniques have been developed toward learning disentangled representations for generative modeling [3, 29]. InfoGAN [3] disentangles hidden dimensions on unlabeled data by mutual information regularization. However, the semantic of each disentangled dimension is uncontrollable because it is discovered after training rather than designated by user modeling. We establish some connections between SGAN and InfoGAN in the next section.

There is also interest in developing DGMs for semi-supervised conditional generation, such as semi-supervised CVAE [11], its many variants [16, 9, 18], ALI [5] and TripleGAN [15], among which the closest to us are [15, 9]. In [9], VAE is enhanced with a discriminator loss and an independency

constraint, and trained via joint MLE and discriminator loss minimization. By contrast, SGAN is an adversarial framework that is trained to match two joint distributions in the visible space, thus avoids MLE for visible variables. TripleGAN builds a three-player adversarial game to drive the generator to match the conditional distribution $p(\boldsymbol{x}|\boldsymbol{y})$, while SGAN models the conditional distribution $p(\boldsymbol{x}|\boldsymbol{y},\boldsymbol{z})$ instead. TripleGAN therefore lacks constraints to ensure the semantics of interest to be exclusively captured by $\boldsymbol{y}$, and lacks a mechanism to perform posterior inference for $\boldsymbol{z}$.

## 3 Structured Generative Adversarial Networks (SGAN)

We build our model based on the generative adversarial networks (GANs) [8], a framework for learning DGMs using a two-player adversarial game. Specifically, given observed data $\{\boldsymbol{x}_i\}_{i=1}^{N}$, GANs try to estimate a generator distribution $p_g(\boldsymbol{x})$ to match the true data distribution $p_{data}(\boldsymbol{x})$, where $p_g(\boldsymbol{x})$ is modeled as a neural network $G$ that transforms a noise variable $\boldsymbol{z} \sim \mathcal{N}(\boldsymbol{0},\boldsymbol{1})$ into generated data $\hat{\boldsymbol{x}} = G(\boldsymbol{z})$. GANs assess the quality of $\hat{\boldsymbol{x}}$ by introducing a neural network discriminator $D$ to judge whether a sample is from $p_{data}(\boldsymbol{x})$ or the generator distribution $p_g(\boldsymbol{x})$. $D$ is trained to distinguish generated samples from true samples while $G$ is trained to fool $D$:

$$\min_{G} \max_{D} \mathcal{L}(D,G) = \mathbb{E}_{\boldsymbol{x} \sim p_{data}(\boldsymbol{x})}[\log(D(\boldsymbol{x}))] + \mathbb{E}_{\boldsymbol{z} \sim p(\boldsymbol{z})}[\log(1 - D(G(\boldsymbol{z})))],$$

Goodfellow et al. [8] show the global optimum of the above problem is attained at $p_g = p_{data}$. It is noted that the original GAN models the latent space using a single unstructured noise variable $\boldsymbol{z}$. The semantics and structures that may be of our interest are entangled in $\boldsymbol{z}$, and the generator transforms $\boldsymbol{z}$ into $\hat{\boldsymbol{x}}$ in a highly uncontrollable way – it lacks both disentanglability and controllability.

We next describe SGAN, a generic extension to GANs that is enhanced with improved disentanglability and controllability for semi-supervised conditional generative modeling.

**Overview.** We consider a semi-supervised setting, where we observe a large set of unlabeled data $\boldsymbol{X} = \{\boldsymbol{x}_i\}_{i=1}^{N}$. We are interested in both the observed sample $\boldsymbol{x}$ and some hidden structures $\boldsymbol{y}$ of $\boldsymbol{x}$, and want to build a conditional generator that can generate data $\hat{\boldsymbol{x}}$ that matches the true data distribution of $\boldsymbol{x}$, while obey the structures specified in $\boldsymbol{y}$ (e.g. generate pictures of digits given 0-9). Besides the unlabeled $\boldsymbol{x}$, we also have access to a small chunk of data $\boldsymbol{X}_l = \{\boldsymbol{x}_j^l, \boldsymbol{y}_j^l\}_{j=1}^{M}$ where the structure $\boldsymbol{y}$ is jointly observed. Therefore, our model needs to characterize the joint distribution $p(\boldsymbol{x},\boldsymbol{y})$ instead of the marginal $p(\boldsymbol{x})$, for both fully and partially observed $\boldsymbol{x}$.

As the data generation process is intrinsically complex and usually determined by many factors beyond $\boldsymbol{y}$, it is necessary to consider other factors that are irrelevant with $\boldsymbol{y}$, and separate the hidden space into two parts $(\boldsymbol{y},\boldsymbol{z})$, of which $\boldsymbol{y}$ encodes the designated semantics, and $\boldsymbol{z}$ includes any other factors of variation [3]. We make a mild assumption that $\boldsymbol{y}$ and $\boldsymbol{z}$ are independent from each other so that $\boldsymbol{y}$ could be disentangled from $\boldsymbol{z}$. Our model thus needs to take into consideration the uncertainty of both $(\boldsymbol{x},\boldsymbol{y})$ and $\boldsymbol{z}$, i.e. characterizing the joint distribution $p(\boldsymbol{x},\boldsymbol{y},\boldsymbol{z})$ while being able to disentangle $\boldsymbol{y}$ from $\boldsymbol{z}$. Directly estimating $p(\boldsymbol{x},\boldsymbol{y},\boldsymbol{z})$ is difficult, as (1) we have never observed $\boldsymbol{z}$ and only observed $\boldsymbol{y}$ for partial $\boldsymbol{x}$; (2) $\boldsymbol{y}$ and $\boldsymbol{z}$ might be entangled at any time as the training proceeds. As an alternative, SGAN builds two inference networks $I$ and $C$. The two inference networks define two distributions $p_i(\boldsymbol{z}|\boldsymbol{x})$ and $p_c(\boldsymbol{y}|\boldsymbol{x})$ that are trained to approximate the true posteriors $p(\boldsymbol{z}|\boldsymbol{x})$ and $p(\boldsymbol{y}|\boldsymbol{x})$ using two different adversarial games. The two games are unified via a shared generator $\boldsymbol{x} \sim p_g(\boldsymbol{x}|\boldsymbol{y},\boldsymbol{z})$. Marginalizing out $\boldsymbol{z}$ or $\boldsymbol{y}$ obtains $p_g(\boldsymbol{x}|\boldsymbol{z})$ and $p_g(\boldsymbol{x}|\boldsymbol{y})$:

$$p_g(\boldsymbol{x}|\boldsymbol{z}) = \int_{\boldsymbol{y}} p(\boldsymbol{y})p_g(\boldsymbol{x}|\boldsymbol{y},\boldsymbol{z})d\boldsymbol{y}, \; p_g(\boldsymbol{x}|\boldsymbol{y}) = \int_{\boldsymbol{z}} p(\boldsymbol{z})p_g(\boldsymbol{x}|\boldsymbol{y},\boldsymbol{z})d\boldsymbol{z}, \tag{1}$$

where $p(\boldsymbol{y})$ and $p(\boldsymbol{z})$ are appropriate known priors for $\boldsymbol{y}$ and $\boldsymbol{z}$. As SGAN is able to perform posterior inference for both $\boldsymbol{z}$ and $\boldsymbol{y}$ given $\boldsymbol{x}$ (even for unlabeled data), we can directly imposes constraints [13] that enforce the structures of interest being exclusively captured by $\boldsymbol{y}$, while those irreverent factors being encoded in $\boldsymbol{z}$ (as we will show later). Fig.1 illustrates the key components of SGAN, which we elaborate as follows.

**Generator $p_g(\boldsymbol{x}|\boldsymbol{y},\boldsymbol{z})$.** We assume the following generative process from $\boldsymbol{y},\boldsymbol{z}$ to $\boldsymbol{x}$: $\boldsymbol{z} \sim p(\boldsymbol{z}), \boldsymbol{y} \sim p(\boldsymbol{y}), \boldsymbol{x} \sim p(\boldsymbol{x}|\boldsymbol{y},\boldsymbol{z})$, where $p(\boldsymbol{z})$ is chosen as a non-informative prior, and $p(\boldsymbol{y})$ as an appropriate prior that meets our modeling needs (e.g. a categorical distribution for digit class). We parametrize $p(\boldsymbol{x}|\boldsymbol{y},\boldsymbol{z})$ using a neural network generator $G$, which takes $\boldsymbol{y}$ and $\boldsymbol{z}$ as inputs, and outputs generated samples $\boldsymbol{x} \sim p_g(\boldsymbol{x}|\boldsymbol{y},\boldsymbol{z}) = G(\boldsymbol{y},\boldsymbol{z})$. $G$ can be seen as a "decoder" in VAE parlance, and its architecture depends on specific applications, such as a deconvolutional neural network for generating images [25, 21].

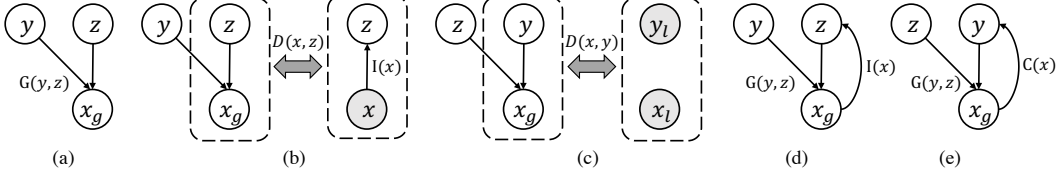

Figure 1: An overview of the SGAN model: (a) the generator $p_g(\boldsymbol{x}|\boldsymbol{y}, \boldsymbol{z})$; (b) the adversarial game $\mathcal{L}_{\boldsymbol{xz}}$; (c) the adversarial game $\mathcal{L}_{\boldsymbol{xy}}$; (d) the collaborative game $\mathcal{R}_{\boldsymbol{z}}$; (e) the collaborative game $\mathcal{R}_{\boldsymbol{y}}$.

**Adversarial game $\mathcal{L}_{\boldsymbol{xz}}$.** Following the adversarially learning inference (ALI) framework, we construct an adversarial game to match the distributions of joint pairs $(\boldsymbol{x}, \boldsymbol{z})$ drawn from the two different factorizations: $p_g(\boldsymbol{x}, \boldsymbol{z}) = p(\boldsymbol{z})p_g(\boldsymbol{x}|\boldsymbol{z})$, $p_i(\boldsymbol{x}, \boldsymbol{z}) = p(\boldsymbol{x})p_i(\boldsymbol{z}|\boldsymbol{x})$. Specifically, to draw samples from $p_g(\boldsymbol{x}, \boldsymbol{z})$, we note the fact that we can first draw the tuple $(\boldsymbol{x}, \boldsymbol{y}, \boldsymbol{z})$ following $\boldsymbol{y} \sim p(\boldsymbol{y}), \boldsymbol{z} \sim p(\boldsymbol{z}), \boldsymbol{x} \sim p_g(\boldsymbol{x}|\boldsymbol{y}, \boldsymbol{z})$, and then only taking $(\boldsymbol{x}, \boldsymbol{z})$ as needed. This implicitly performs the marginalization as in Eq. 1. On the other hand, we introduce an *inference network* $I : \boldsymbol{x} \to \boldsymbol{z}$ to approximate the true posterior $p(\boldsymbol{z}|\boldsymbol{x})$. Obtaining $(\boldsymbol{x}, \boldsymbol{z}) \sim p(\boldsymbol{x})p_i(\boldsymbol{z}|\boldsymbol{x})$ with $I$ is straightforward: $\boldsymbol{x} \sim p(\boldsymbol{x}), \boldsymbol{z} \sim p_i(\boldsymbol{z}|\boldsymbol{x}) = I(\boldsymbol{x})$. Training $G$ and $I$ involves finding the Nash equilibrium for the following minimax game $\mathcal{L}_{\boldsymbol{xz}}$ (we slightly abuse $\mathcal{L}_{\boldsymbol{xz}}$ for both the minimax objective and a name for this adversarial game):

$$\min_{I,G} \max_{D_{\boldsymbol{xz}}} \mathcal{L}_{\boldsymbol{xz}} = \mathbb{E}_{\boldsymbol{x} \sim p(\boldsymbol{x})}[\log(D_{\boldsymbol{xz}}(\boldsymbol{x}, I(\boldsymbol{x})))] + \mathbb{E}_{\boldsymbol{z} \sim p(\boldsymbol{z}), \boldsymbol{y} \sim p(\boldsymbol{y})}[\log(1 - D_{\boldsymbol{xz}}(G(\boldsymbol{y}, \boldsymbol{z}), \boldsymbol{z}))], \quad (2)$$

where we introduce $D_{\boldsymbol{xz}}$ as a critic network that is trained to distinguish pairs $(\boldsymbol{x}, \boldsymbol{z}) \sim p_g(\boldsymbol{x}, \boldsymbol{z})$ from those come from $p_i(\boldsymbol{x}, \boldsymbol{z})$. This minimax objective reaches optimum if and only if the conditional distribution $p_g(\boldsymbol{x}|\boldsymbol{z})$ characterized by $G$ inverses the approximate posterior $p_i(\boldsymbol{z}|\boldsymbol{x})$, implying $p_g(\boldsymbol{x}, \boldsymbol{z}) = p_i(\boldsymbol{x}, \boldsymbol{z})$ [4, 5]. As we have never observed $\boldsymbol{z}$ for $\boldsymbol{x}$, as long as $\boldsymbol{z}$ is assumed to be independent from $\boldsymbol{y}$, it is reasonable to just set the true joint distribution $p(\boldsymbol{x}, \boldsymbol{z}) = p_g^*(\boldsymbol{x}, \boldsymbol{z}) = p_i^*(\boldsymbol{x}, \boldsymbol{z})$, where we use $p_g^*$ and $p_i^*$ to denote the optimal distributions when $\mathcal{L}_{\boldsymbol{xz}}$ reaches its equilibrium.

**Adversarial game $\mathcal{L}_{\boldsymbol{xy}}$.** The second adversarial game is built to match the true joint data distribution $p(\boldsymbol{x}, \boldsymbol{y})$ that has been observed on $\boldsymbol{X}_l$. We introduce the other critic network $D_{\boldsymbol{xy}}$ to discriminate $(\boldsymbol{x}, \boldsymbol{y}) \sim p(\boldsymbol{x}, \boldsymbol{y})$ from $(\boldsymbol{x}, \boldsymbol{y}) \sim p_g(\boldsymbol{x}, \boldsymbol{y}) = p(\boldsymbol{y})p_g(\boldsymbol{x}|\boldsymbol{y})$, and build the game $\mathcal{L}_{\boldsymbol{xy}}$ as:

$$\min_{G} \max_{D_{\boldsymbol{xy}}} \mathcal{L}_{\boldsymbol{xy}} = \mathbb{E}_{(\boldsymbol{x}, \boldsymbol{y}) \sim p(\boldsymbol{x}, \boldsymbol{y})}[\log(D_{\boldsymbol{xy}}(\boldsymbol{x}, \boldsymbol{y}))] + \mathbb{E}_{\boldsymbol{y} \sim p(\boldsymbol{y}), \boldsymbol{z} \sim p(\boldsymbol{z})}[\log(1 - D_{\boldsymbol{xy}}(G(\boldsymbol{y}, \boldsymbol{z}), \boldsymbol{y}))].$$

$$(3)$$

**Collaborative game $\mathcal{R}_{\boldsymbol{y}}$.** Although training the adversarial game $\mathcal{L}_{\boldsymbol{xy}}$ theoretically drives $p_g(\boldsymbol{x}, \boldsymbol{y})$ to concentrate on the true data distribution $p(\boldsymbol{x}, \boldsymbol{y})$, it turns out to be very difficult to train $\mathcal{L}_{\boldsymbol{xy}}$ to desired convergence, as (1) the joint distribution $p(\boldsymbol{x}, \boldsymbol{y})$ characterized by $\boldsymbol{X}_l$ might be biased due to its small data size; (2) there is little supervision from $\boldsymbol{X}_l$ to tell $G$ what $\boldsymbol{y}$ essentially represents, and how to generate samples conditioned on $\boldsymbol{y}$. As a result, $G$ might lack controllability – it might generate low-fidelity samples that are not aligned with their conditions, which will always be rejected by $D_{\boldsymbol{xy}}$. A natural solution to these issues is to allow (learned) posterior inference of $\boldsymbol{y}$ to reconstruct $\boldsymbol{y}$ from generated $\boldsymbol{x}$ [5]. By minimizing the reconstruction error, we can backpropagate the gradient to $G$ to enhance its controllability. Once $p_g(\boldsymbol{x}|\boldsymbol{y})$ can generate high-fidelity samples that respect the structures $\boldsymbol{y}$, we can reuse the generated samples $(\boldsymbol{x}, \boldsymbol{y}) \sim p_g(\boldsymbol{x}, \boldsymbol{y})$ as true samples in the first term of $\mathcal{L}_{\boldsymbol{xy}}$, to prevent $D_{\boldsymbol{xz}}$ from collapsing into a biased $p(\boldsymbol{x}, \boldsymbol{y})$ characterized by $\boldsymbol{X}_l$.

Intuitively, we introduce the second inference network $C : \boldsymbol{x} \to \boldsymbol{y}$ which approximates the posterior $p(\boldsymbol{y}|\boldsymbol{x})$ as $\boldsymbol{y} \sim p_c(\boldsymbol{y}|\boldsymbol{x}) = C(\boldsymbol{x})$, e.g. $C$ reduces to a N-way classifier if $\boldsymbol{y}$ is categorical. To train $p_c(\boldsymbol{y}|\boldsymbol{x})$, we define a collaboration (reconstruction) game $\mathcal{R}_{\boldsymbol{y}}$ in the hidden space of $\boldsymbol{y}$:

$$\min_{C,G} \mathcal{R}_{\boldsymbol{y}} = -\mathbb{E}_{(\boldsymbol{x}, \boldsymbol{y}) \sim p(\boldsymbol{x}, \boldsymbol{y})}[\log p_c(\boldsymbol{y}|\boldsymbol{x})] - \mathbb{E}_{(\boldsymbol{x}, \boldsymbol{y}) \sim p_g(\boldsymbol{x}, \boldsymbol{y})}[\log p_c(\boldsymbol{y}|\boldsymbol{x})], \quad (4)$$

which aims to minimize the reconstruction error of $\boldsymbol{y}$ in terms of $C$ and $G$, on both labeled data $\boldsymbol{X}_l$ and generated data $(\boldsymbol{x}, \boldsymbol{y}) \sim p_g(\boldsymbol{x}, \boldsymbol{y})$. On the one hand, minimizing the first term of $\mathcal{R}_{\boldsymbol{y}}$ w.r.t. $C$ guides $C$ toward the true posterior $p(\boldsymbol{y}|\boldsymbol{x})$. On the other hand, minimizing the second term w.r.t. $G$ enhances $G$ with extra controllability – it minimizes the chance that $G$ could generate samples that would otherwise be falsely predicted by $C$. Note that we also minimize the second term w.r.t. $C$, which proves effective in semi-supervised learning settings that uses synthetic samples to augment the predictive power of $C$. In summary, minimizing $\mathcal{R}_{\boldsymbol{y}}$ can be seen as a collaborative game between two players $C$ and $G$ that drives $p_g(\boldsymbol{x}|\boldsymbol{y})$ to match $p(\boldsymbol{x}|\boldsymbol{y})$ and $p_c(\boldsymbol{y}|\boldsymbol{x})$ to match the posterior $p(\boldsymbol{y}|\boldsymbol{x})$.

**Collaborative games** $\mathcal{R}_{\boldsymbol{z}}$**.** As SGAN allows posterior inference for both $\boldsymbol{y}$ and $\boldsymbol{z}$, we can explicitly impose constraints $\mathcal{R}_{\boldsymbol{y}}$ and $\mathcal{R}_{\boldsymbol{z}}$ to separate $\boldsymbol{y}$ from $\boldsymbol{z}$ during training. To explain, we first note that optimizing the second term of $\mathcal{R}_{\boldsymbol{y}}$ w.r.t $G$ actually enforces the structure information to be fully persevered in $\boldsymbol{y}$, because $C$ is asked to recover the structure $\boldsymbol{y}$ from $G(\boldsymbol{y}, \boldsymbol{z})$, which is generated conditioned on $\boldsymbol{y}$, regardless of the uncertainty of $\boldsymbol{z}$ (as $\boldsymbol{z}$ is marginalized out during sampling). Therefore, minimizing $\mathcal{R}_{\boldsymbol{y}}$ indicates the following constraint: $\min_{C,G} \mathbb{E}_{\boldsymbol{y}\sim p(\boldsymbol{y})}\big\|p_c(\boldsymbol{y}|G(\boldsymbol{y}, \boldsymbol{z}_1)), p_c(\boldsymbol{y}|G(\boldsymbol{y}, \boldsymbol{z}_2))\big\|, \forall \boldsymbol{z}_1, \boldsymbol{z}_2 \sim p(\boldsymbol{z})$, where $\|\boldsymbol{a}, \boldsymbol{b}\|$ is some distance function between $\boldsymbol{a}$ and $\boldsymbol{b}$ (e.g. cross entropy if $C$ is a N-way classifier). On the counter part, we also want to enforce any other unstructured information that is not of our interest to be fully captured in $\boldsymbol{z}$, without being entangled with $\boldsymbol{y}$. So we build the second collaborative game $\mathcal{R}_{\boldsymbol{z}}$ as:

$$\min_{I,G} \mathcal{R}_{\boldsymbol{z}} = -\mathbb{E}_{(\boldsymbol{x},\boldsymbol{z})\sim p_g(\boldsymbol{x},\boldsymbol{z})}[\log p_i(\boldsymbol{z}|\boldsymbol{x})] \tag{5}$$

where $I$ is required to recover $\boldsymbol{z}$ from those samples generated by $G$ conditioned on $\boldsymbol{z}$, i.e. reconstructing $\boldsymbol{z}$ in the hidden space. Similar to $\mathcal{R}_{\boldsymbol{y}}$, minimizing $\mathcal{R}_{\boldsymbol{z}}$ indicates: $\min_{I,G} \mathbb{E}_{\boldsymbol{z}\sim p(\boldsymbol{z})}\big\|p_i(\boldsymbol{z}|G(\boldsymbol{y}_1, \boldsymbol{z})), p_i(\boldsymbol{z}|G(\boldsymbol{y}_2, \boldsymbol{z}))\big\|, \forall \boldsymbol{y}_1, \boldsymbol{y}_2 \sim p(\boldsymbol{y})$, and when we model $I$ as a deterministic mapping [4], the $\|\cdot\|$ distance between distributions is equal to the $\ell$-2 distance between the outputs of $I$.

**Theoretical Guarantees.** We provide some theoretical results about the SGAN framework under the nonparametric assumption. The proofs of the theorems are deferred to the supplementary materials.

**Theorem 3.1** *The global minimum of* $\max_{D_{\boldsymbol{xz}}} \mathcal{L}_{\boldsymbol{xz}}$ *is achieved if and only if* $p(\boldsymbol{x})p_i(\boldsymbol{z}|\boldsymbol{x}) = p(\boldsymbol{z})p_g(\boldsymbol{x}|\boldsymbol{z})$. *At that point* $D_{\boldsymbol{xz}}^* = \frac{1}{2}$. *Similarly, the global minimum of* $\max_{D_{\boldsymbol{xy}}} \mathcal{L}_{\boldsymbol{xy}}$ *is achieved if and only if* $p(\boldsymbol{x}, \boldsymbol{y}) = p(\boldsymbol{y})p_g(\boldsymbol{x}|\boldsymbol{y})$. *At that point* $D_{\boldsymbol{xy}}^* = \frac{1}{2}$.

**Theorem 3.2** *There exists a generator* $G^*(\boldsymbol{y}, \boldsymbol{z})$ *of which the conditional distributions* $p_g(\boldsymbol{x}|\boldsymbol{y})$ *and* $p_g(\boldsymbol{x}|\boldsymbol{z})$ *can both achieve equilibrium in their own minimax games* $\mathcal{L}_{\boldsymbol{xy}}$ *and* $\mathcal{L}_{\boldsymbol{xz}}$.

**Theorem 3.3** *Minimizing* $\mathcal{R}_{\boldsymbol{z}}$ *w.r.t. $I$ will keep the equilibrium of the adversarial game* $\mathcal{L}_{\boldsymbol{xz}}$. *Similarly, minimizing* $\mathcal{R}_{\boldsymbol{y}}$ *w.r.t. $C$ will keep the equilibrium of the adversarial game* $\mathcal{L}_{\boldsymbol{xy}}$ *unchanged.*

---

**Algorithm 1** Training Structured Generative Adversarial Networks (SGAN).

---

1: Pretrain $C$ by minimizing the first term of Eq. 4 w.r.t. $C$ using $\boldsymbol{X}_l$.
2: **repeat**
3:     Sample a batch of $\boldsymbol{x}$: $\boldsymbol{x}_u \sim p(\boldsymbol{x})$.
4:     Sample batches of pairs $(\boldsymbol{x}, \boldsymbol{y})$: $(\boldsymbol{x}_l, \boldsymbol{y}_l) \sim p(\boldsymbol{x}, \boldsymbol{y})$, $(\boldsymbol{x}_g, \boldsymbol{y}_g) \sim p_g(\boldsymbol{x}, \boldsymbol{y})$, $(\boldsymbol{x}_c, \boldsymbol{y}_c) \sim p_c(\boldsymbol{x}, \boldsymbol{y})$.
5:     Obtain a batch $(\boldsymbol{x}_m, \boldsymbol{y}_m)$ by mixing data from $(\boldsymbol{x}_l, \boldsymbol{y}_l), (\boldsymbol{x}_g, \boldsymbol{y}_g), (\boldsymbol{x}_c, \boldsymbol{y}_c)$ with proper mixing portion.
6:     **for** $k = 1 \rightarrow K$ **do**
7:         Train $D_{\boldsymbol{xz}}$ by maximizing the first term of $\mathcal{L}_{\boldsymbol{xz}}$ using $\boldsymbol{x}_u$ and the second using $\boldsymbol{x}_g$.
8:         Train $D_{\boldsymbol{xy}}$ by maximizing the first term of $\mathcal{L}_{\boldsymbol{xy}}$ using $(\boldsymbol{x}_m, \boldsymbol{y}_m)$ and the second using $(\boldsymbol{x}_g, \boldsymbol{y}_g)$.
9:     **end for**
10:    Train $I$ by minimizing $\mathcal{L}_{\boldsymbol{xz}}$ using $\boldsymbol{x}_u$ and $\mathcal{R}_{\boldsymbol{z}}$ using $\boldsymbol{x}_g$.
11:    Train $C$ by minimizing $\mathcal{R}_{\boldsymbol{y}}$ using $(\boldsymbol{x}_m, \boldsymbol{y}_m)$ (see text).
12:    Train $G$ by minimizing $\mathcal{L}_{\boldsymbol{xy}} + \mathcal{L}_{\boldsymbol{xz}} + \mathcal{R}_{\boldsymbol{y}} + \mathcal{R}_{\boldsymbol{z}}$ using $(\boldsymbol{x}_g, \boldsymbol{y}_g)$.
13: **until** convergence.

---

**Training.** SGAN is fully differentiable and can be trained end-to-end using stochastic gradient descent, following the strategy in [8] that alternatively trains the two critic networks $D_{\boldsymbol{xy}}, D_{\boldsymbol{xz}}$ and the other networks $G$, $I$ and $C$. Though minimizing $\mathcal{R}_{\boldsymbol{y}}$ and $\mathcal{R}_{\boldsymbol{z}}$ w.r.t. $G$ will introduce slight bias, we find empirically it works well and contributes to disentangling $\boldsymbol{y}$ and $\boldsymbol{z}$. The training procedures are summarized in Algorithm 1. Moreover, to guarantee that $C$ could be properly trained without bias, we pretrain $C$ by minimizing the first term of $\mathcal{R}_{\boldsymbol{y}}$ until convergence, and do not minimize $\mathcal{R}_{\boldsymbol{y}}$ w.r.t. $C$ until $G$ has started generating meaning samples (usually after several epochs of training). As the training proceeds, we gradually improve the portion of synthetic samples $(\boldsymbol{x}, \boldsymbol{y}) \sim p_g(\boldsymbol{x}, \boldsymbol{y})$ and $(\boldsymbol{x}, \boldsymbol{y}) \sim p_c(\boldsymbol{x}, \boldsymbol{y})$ in the stochastic batch, to help the training of $D_{\boldsymbol{xy}}$ and $C$ (see Algorithm 1), and you can refer to our codes on GitHub for more details of the portion. We empirically found this mutual bootstrapping trick yields improved $C$ and $G$.

**Discussion and connections.** SGAN is essentially a combination of two adversarial games $\mathcal{L}_{\boldsymbol{xy}}$ and $\mathcal{L}_{\boldsymbol{xz}}$, and two collaborative games $\mathcal{R}_{\boldsymbol{y}}, \mathcal{R}_{\boldsymbol{z}}$, where $\mathcal{L}_{\boldsymbol{xy}}$ and $\mathcal{L}_{\boldsymbol{xz}}$ are optimized to match the data distributions in the visible space, while $\mathcal{R}_{\boldsymbol{y}}$ and $\mathcal{R}_{\boldsymbol{z}}$ are trained to match the posteriors in the hidden space. It combines the best of GAN-based methods and MLE-based methods: on one hand, estimating

density in the visible space using GAN-based formulation avoids distributing the probability mass diffusely over data space [5], which MLE-based frameworks (e.g. VAE) suffer. One the other hand, incorporating reconstruction-based constraints in latent space helps enforce the disentanglement between structured information in $\boldsymbol{y}$ and unstructured ones in $\boldsymbol{z}$, as we argued above.

We also establish some connections between SGAN and some existing works [15, 27, 3]. We note the $\mathcal{L}_{\boldsymbol{xy}}$ game in SGAN is connected to the TripleGAN framework [15] when its trade-off parameter $\alpha = 0$. We will empirically show that SGAN yields better controllability on $G$, and also improved performance on downstream tasks, due to the separate modeling of $\boldsymbol{y}$ and $\boldsymbol{z}$. SGAN also connects to InfoGAN in the sense that the second term of $\mathcal{R}_{\boldsymbol{y}}$ (Eq. 4) reduces to the mutual information penalty in InfoGAN under unsupervised settings. However, SGAN and InfoGAN have totally different aims and modeling techniques. SGAN builds a conditional generator that has the semantic of interest $\boldsymbol{y}$ as a fully controllable input (known before training); InfoGAN in contrast aims to disentangle some latent variables whose semantics are interpreted after training (by observation). Though extending InfoGAN to semi-supervised settings seems straightforward, successfully learning the joint distribution $p(\boldsymbol{x}, \boldsymbol{y})$ with very few labels is non-trivial: InfoGAN only maximizes the mutual information between $\boldsymbol{y}$ and $G(\boldsymbol{y}, \boldsymbol{z})$, bypassing $p(\boldsymbol{y}|\boldsymbol{x})$ or $p(\boldsymbol{x}, \boldsymbol{y})$, thus its direct extension to semi-supervised settings may fail due to lack of $p(\boldsymbol{x}, \boldsymbol{y})$. Moreover, SGAN has dedicated inference networks $I$ and $C$, while the network $Q(\boldsymbol{x})$ in InfoGAN shares parameters with the discriminator, which has been argued as problematic [15, 9] as it may compete with the discriminator and prevents its success in semi-supervised settings. See our ablation study in section 4.2 and Fig.3. Finally, the first term in $\mathcal{R}_{\boldsymbol{y}}$ is similar to the way Improved-GAN models the conditional $p(\boldsymbol{y}|\boldsymbol{x})$ for labeled data, but SGAN treats the generated data very differently – Improved-GAN labels $\boldsymbol{x}_g = G(\boldsymbol{z}, \boldsymbol{y})$ as a new class $\boldsymbol{y} = K + 1$, instead SGAN reuses $\boldsymbol{x}_g$ and $\boldsymbol{x}_c$ to mutually boost $I$, $C$ and $G$, which is key to the success of semi-supervised learning (see section 4.2).

# 4  Evaluation

We empirically evaluate SGAN through experiments on different datasets. We show that separately modeling $\boldsymbol{z}$ and $\boldsymbol{y}$ in the hidden space helps better disentangle the semantics of our interest from other irrelevant attributes, thus yields improved performance for both generative modeling ($G$) and posterior inference ($C$, $I$) (section 4.1 4.3). Under SGAN framework, the learned inference networks and generators can further benefit a lot of downstream applications, such as semi-supervised classification, controllable image generation and style transfer (section 4.2 4.3).

**Dataset and configurations.** We evaluate SGAN on three image datasets: (1) `MNIST` [14]: we use the 60K training images as unlabeled data, and sample $n \in \{20, 50, 100\}$ labels for semi-supervised learning following [12, 27], and evaluate on the 10K test images. (2) `SVHN` [20]: a standard train/test split is provided, where we sample $n = 1000$ labels from the training set for semi-supervised learning [27, 15, 5]. (3) `CIFAR-10`: a challenging dataset for conditional image generation that consists of 50K training and 10K test images from 10 object classes. We randomly sample $n = 4000$ labels [27, 28, 15] for semi-supervised learning. For all datasets, our semantic of interest is the digit/object class, so $\boldsymbol{y}$ is a 10-dim categorical variable. We use a 64-dim gaussian noise as $\boldsymbol{z}$ in MNIST and a 100-dim uniform noise as $\boldsymbol{z}$ in SVHN and CIFAR-10.

**Implementation.** We implement SGAN using TensorFlow [1] and Theano [2] with distributed acceleration provided by Poseidon [33] which parallelizes line 7-8 and 10-12 of Algorithm. 1. The neural network architectures of $C$, $G$ and $D_{xy}$ mostly follow those used in TripleGAN [15] and we design $I$ and $D_{xz}$ according to [5] but with shallower structures to alleviate the training costs. Empirically SGAN needs 1.3-1.5x more training time than TripleGAN [15] without parallelization. It is noted that properly weighting the losses of the four games in SGAN during training may lead to performance improvement. However, we simply set them equal without heavy tuning[1].

## 4.1  Controllability and Disentanglability

We evaluate the controllability and disentanglability of SGAN by assessing its generator network $G$ and inference network $I$, respectively. Specifically, as SGAN is able to perform posterior inference for $\boldsymbol{z}$, we define a novel quantitative measure based on $\boldsymbol{z}$ to compare its disentanglability to other DGMs: we first use the trained $I$ (or the "recognition network" in VAE-based models) to infer $\boldsymbol{z}$ for unseen $\boldsymbol{x}$ from test sets. Ideally, as $\boldsymbol{z}$ and $\boldsymbol{y}$ are modeled as independent, when $I$ is trained to approach the true posterior of $\boldsymbol{z}$, its output, when used as features, shall have weak predictability for $\boldsymbol{y}$. Accordingly, we

use $z$ as features to train a linear SVM classifier to predict the true $y$, and define the converged accuracy of this classifier as the *mutual predictability (MP)* measure, and expect lower MP for models that can better disentangle $y$ from $z$. We conduct this experiment on all three sets, and report the averaged MP measure of five runs in Fig. 2, comparing the following DGMs (that are able to infer $z$): (1) `ALI` [5] and (2) `VAE` [12], trained without label information; (3) `CVAE-full`[2]: the M2 model in [11] trained under the fully supervised setting; (4) `SGAN` trained under semi-supervised settings. We use 50, 1000 and 4000 labels for MNIST, SVHN and CIFAR-10 dataset under semi-supervised settings, respectively.

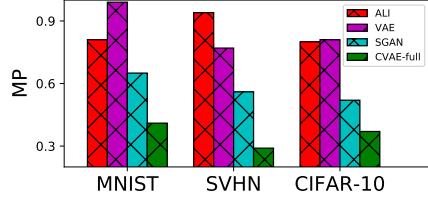

Clearly, SGAN demonstrates low MP when predicting $y$ using $z$ on three datasets. Using only 50 labels, SGAN exhibits reasonable MP. In fact, on MNIST with only 20 labels as supervision, SGAN achieves $0.65$ MP, outperforming other baselines by a large margin. The results clearly demonstrate SGAN's ability to disentangle $y$ and $z$, even when the supervision is very scarce.

Figure 2: Comparisons of the MP measure for different DGMs (lower is better).

On the other hand, better disentanglability also implies improved controllability of $G$, because less entangled $y$ and $z$ would be easier for $G$ to recognize the designated semantics – so $G$ should be able to generate samples that are less deviated from $y$ during conditional generation. To verify this, following [9], we use a pretrained gold-standard classifier ($0.56\%$ error on MNIST test set) to classify generated images, and use the condition $y$ as ground truth to calculate the accuracy. We compare SGAN in Table 1 to `CVAE-semi` and `TripleGAN` [15], another strong baseline that is also designed for conditional generation under semi-supervised settings. We use $n = 20, 50, 100$ labels on MNIST, and observe a significantly higher accuracy for both `TripleGAN` and `SGAN`. For comparison, a generator trained by `CVAE-full` achieves $0.6\%$ error. When there are fewer labels available, SGAN outperforms `TripleGAN`. The generator in SGAN can generate samples that consistently obey the conditions specified in $y$, even when there are only two images per class ($n = 20$) as supervision. These results verify our statements that disentangled semantics further enhance the controllability of the conditioned generator $G$.

## 4.2 Semi-supervised Classification

It is natural to use SGAN for semi-supervised prediction. With a little supervision, SGAN can deliver a conditional generator with reasonably good controllability, with which, one can synthesize samples from $p_g(\boldsymbol{x}, \boldsymbol{y})$ to augment the training of $C$ when minimizing $\mathcal{R}_{\boldsymbol{y}}$. Once $C$ becomes more accurate, it tends to make less

| Model | # labeled samples | | |
|---|---|---|---|
| | $n = 20$ | $n = 50$ | $n = 100$ |
| `CVAE-semi` | 33.05 | 10.72 | 5.66 |
| `TripleGAN` | 3.06 | 1.80 | 1.29 |
| `SGAN` | **1.68** | **1.23** | **0.93** |

Table 1: Errors (%) of generated samples classified by a classifier with $0.56\%$ test error.

mistakes when inferring $y$ from $x$. Moreover, as we are sampling $(\boldsymbol{x}, \boldsymbol{y}) \sim p_c(\boldsymbol{x}, \boldsymbol{y})$ to train $D_{\boldsymbol{xy}}$ during the maximization of $\mathcal{L}_{\boldsymbol{xy}}$, a more accurate $C$ means more available labeled samples (by predicting $y$ from unlabeled $x$ using $C$) to lower the bias brought by the small set $\boldsymbol{X}_l$, which in return can enhance $G$ in the minimization phase of $\mathcal{L}_{\boldsymbol{xy}}$. Consequently, a mutual boosting cycle between $G$ and $C$ is formed.

To empirically validate this, we deploy SGAN for semi-supervised classification on MNIST, SVHN and CIFAR-10, and compare the test errors of $C$ to strong baselines in Table 2. To keep the comparisons fair, we adopt the same neural network architectures and hyper-parameter settings from [15], and report the averaged results of 10 runs with randomly sampled labels (every class has equal number of labels). We note that SGAN outperforms the current state-of-the-art methods across all datasets and settings. Especially, on MNIST when labeled instances are very scarce ($n = 20$), SGAN attains the highest accuracy (4.0% test error) with significantly lower variance, benefiting from the mutual boosting effects explained above. This is very critical for applications under low-shot or even one-shot settings where the small set $\boldsymbol{X}_l$ might not be a good representative for the data distribution $p(\boldsymbol{x}, \boldsymbol{y})$.

| Method | MNIST | | | SVHN | CIFAR-10 |
|---|---|---|---|---|---|
| | $n = 20$ | $n = 50$ | $n = 100$ | $n = 1000$ | $n = 4000$ |
| Ladder [22] | - | - | **0.89($\pm$0.50)** | - | 20.40($\pm$0.47) |
| VAE [12] | - | - | 3.33($\pm$0.14) | 36.02($\pm$0.10) | - |
| CatGAN [28] | - | - | 1.39($\pm$0.28) | - | 19.58($\pm$0.58) |
| ALI [5] | - | - | - | 7.3 | 18.3 |
| ImprovedGAN [27] | 16.77($\pm$4.52) | 2.21($\pm$1.36) | 0.93 ($\pm$0.07) | 8.11($\pm$1.3) | 18.63($\pm$2.32) |
| TripleGAN [15] | 5.40($\pm$6.53) | 1.59($\pm$0.69) | 0.92($\pm$0.58) | 5.83($\pm$0.20) | 18.82($\pm$0.32) |
| SGAN | **4.0($\pm$4.14)** | **1.29($\pm$0.47)** | **0.89($\pm$0.11)** | **5.73($\pm$0.12)** | **17.26($\pm$0.69)** |

Table 2: Comparisons of semi-supervised classification errors (%) on MNIST, SVHN and CIFAR-10 test sets.

### 4.3 Qualitative Results

In this section we present qualitative results produced by SGAN's generator under semi-supervised settings. Unless otherwise specified, we use 50, 1000 and 4000 labels on MNIST, SVHN, CIFAR-10 for the results. These results are randomly selected without cherry pick, and more results could be found in the supplementary materials.

**Controllable generation.** To figure out how each module in SGAN contributes to the final results, we conduct an ablation study in Fig.3, where we plot images generated by SGAN with or without the terms $\mathcal{R}_y$ and $\mathcal{R}_z$ during training. As we have observed, our full model accurately disentangles $y$ and $z$. When there is no collaborative game involved, the generator easily collapses to a biased conditional distribution defined by the classifier $C$ that is trained only on a very



(a) w/o $\mathcal{R}_y, \mathcal{R}_z$    (b) w/o $\mathcal{R}_z$    (c) Full model

Figure 3: Ablation study: conditional generation results by SGAN (a) without $\mathcal{R}_y, \mathcal{R}_z$, (b) without $\mathcal{R}_z$ (c) full model. Each row has the same $y$ while each column shares the same $z$.

small set of labeled data with insufficient supervision. For example, the generator cannot clearly distinguish the following digits: 0, 2, 3, 5, 8. Incorporating $\mathcal{R}_y$ into training significantly alleviate this issue – an augmented $C$ would resolve $G$'s confusion. However, it still makes mistakes in some confusing classes, such as 3 and 5. $\mathcal{R}_y$ and $\mathcal{R}_z$ connect the two adversarial games to form a mutual boosting cycle. The absence of any of them would break this cycle, consequently, SGAN would be under-constrained and may collapse to some local minima – resulting in both a less accurate classifier $C$ and a less controlled $G$.

**Visual quality.** Next, we investigate whether a more disentangled $y, z$ will result in higher visual quality on generated samples, as it makes sense that the conditioned generator $G$ would be much easier to learn when its inputs $y$ and $z$ carry more orthogonal information. We conduct this experiment on CIFAR-10 that is consisted of natural images with more uncertainty besides the object categories. We compare several state-of-the-art generators in Fig 4 to SGAN without any advanced GAN training strategies (e.g. WGAN, gradient penalties) that are reported to possibly improve the visual quality. We find SGAN's conditional generator does generate less blurred images with the main objects more salient, compared to TripleGAN and ImprovedGAN w/o minibatch discrimination (see supplementary). For a quantitative measure, we generate 50K images and compute the inception

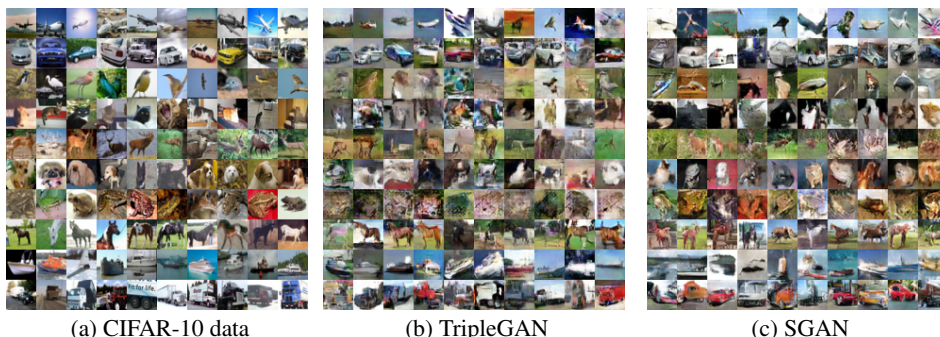

(a) CIFAR-10 data     (b) TripleGAN     (c) SGAN

Figure 4: Visual comparison of generated images on CIFAR-10. For (b) and (c), each row shares the same $y$.

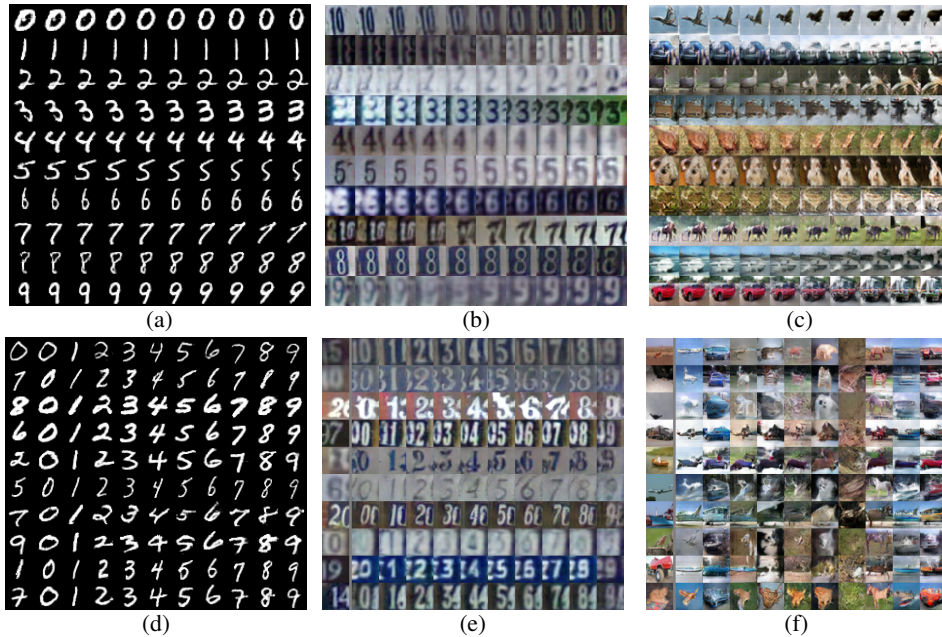

Figure 5: (a)-(c): image progression, (d)-(f): style transfer using SGAN.

score [27] as 6.91(±0.07), compared to `TripleGAN` 5.08(±0.09) and `Improved-GAN` 3.87(±0.03) w/o minibatch discrimination, confirming the advantage of structured modeling for $y$ and $z$.

**Image progression.** To demonstrate that SGAN generalizes well instead of just memorizing the data, we generate images with interpolated $z$ in Fig.5(a)-(c) [32]. Clearly, the images generated with progression are semantically consistent with $y$, and change smoothly from left to right. This verifies that SGAN correctly disentangles semantics, and learns accurate class-conditional distributions.

**Style transfer.** We apply SGAN for style transfer [7, 30]. Specifically, as $y$ is modeled as digit/object category on all three dataset, we suppose $z$ shall encode any other information that are orthogonal to $y$ (probably style information). To see whether $I$ behaves properly, we use SGAN to transfer the unstructured information from $z$ in Fig.5(d)-(f): given an image $x$ (the leftmost image), we infer its unstructured code $z$. We generate images conditioned on $z$, but with different $y$. It is interesting to see that $z$ encodes various aspects of the images, such as the shape, texture, orientation, background information, etc, as expected. Moreover, $G$ can correctly transfer these information to other classes.

## 5    Conclusion

We have presented SGAN for semi-supervised conditional generative modeling, which learns from a small set of labeled instances to disentangle the semantics of our interest from other elements in the latent space. We show that SGAN has improved disentanglability and controllability compared to baseline frameworks. SGAN's design is beneficial to a lot of downstream applications: it establishes new state-of-the-art results on semi-supervised classification, and outperforms strong baseline in terms of the visual quality and inception score on controllable image generation.

## Acknowledgements

Zhijie Deng and Jun Zhu are supported by NSF China (Nos. 61620106010, 61621136008, 61332007), the MIIT Grant of Int. Man. Comp. Stan (No. 2016ZXFB00001), Tsinghua Tiangong Institute for Intelligent Computing and the NVIDIA NVAIL Program. Hao Zhang is supported by the AFRL/DARPA project FA872105C0003. Xiaodan Liang is supported by award FA870215D0002.

## Footnotes

[1]The code is publicly available at `https://github.com/thudzj/StructuredGAN`.

[2]For `CVAE-full`, we use test images and ground truth labels together to infer $z$ when calculating MP. We are unable to compare to semi-supervised `CVAE` as in CVAE inferring $z$ for test images requires image labels as input, which is unfair to other methods.

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
