[Supplementary Material]

# Supplementary Material:
# Structured Generative Adversarial Networks

**[1]Zhijie Deng**[*], **[2,3]Hao Zhang**[*], **[2]Xiaodan Liang**, **[2]Luona Yang**,
**[1,2]Shizhen Xu**, **[1]Jun Zhu**[†], **[3]Eric P. Xing**
[1]Tsinghua University, [2]Carnegie Mellon University, [3]Petuum Inc.
{dzj17,xsz12}@mails.tsinghua.edu.cn, {hao,xiaodan1,luonay1}@cs.cmu.edu,
dcszj@mail.tsinghua.edu.cn, epxing@cs.cmu.edu

## A    Theoretical Analysis

In this section, we provide some theoretical analysis about the SGAN framework under the nonparametric assumption.

**Lemma A.1** *Given two fixed distributions $P(\boldsymbol{x}), Q(\boldsymbol{x})$, the function $f(D(\boldsymbol{x})) = \mathbb{E}_P[\log D(\boldsymbol{x})] + \mathbb{E}_Q[\log(1 - D(\boldsymbol{x}))]$ achieves its maximum $\max_D f(D(\boldsymbol{x}))$ at $D^*(\boldsymbol{x}) = \frac{P(\boldsymbol{x})}{P(\boldsymbol{x})+Q(\boldsymbol{x})}$.*

**Lemma A.2** *The global minimum of the function $g(P(\boldsymbol{x}), Q(\boldsymbol{x})) = \max_D f(D(\boldsymbol{x}))$ is achieved if and only if $P(x) = Q(x)$. At that point $g(P(\boldsymbol{x}), Q(\boldsymbol{x})) = -\log(4)$ and $D^*(\boldsymbol{x}) = \frac{1}{2}$.*

**Theorem A.3** *The global minimum of $\max_{D_{\boldsymbol{xz}}} \mathcal{L}_{\boldsymbol{xz}}$ is achieved if and only if $p(\boldsymbol{x})p_i(\boldsymbol{z}|\boldsymbol{x}) = p(\boldsymbol{z})p_g(\boldsymbol{x}|\boldsymbol{z})$. At that point $D^*_{\boldsymbol{xz}} = \frac{1}{2}$. Similarly, the global minimum of $\max_{D_{\boldsymbol{xy}}} \mathcal{L}_{\boldsymbol{xy}}$ is achieved if and only if $p(\boldsymbol{x}, \boldsymbol{y}) = p(\boldsymbol{y})p_g(\boldsymbol{x}|\boldsymbol{y})$. At that point $D^*_{\boldsymbol{xy}} = \frac{1}{2}$.*

Proofs of Lemma A.1 and A.2 can be found in [2]. Theorem A.3 is a trivial extension of Lemma A.1 and A.2, it shows that the equilibrium of the two games are reached when the conditionals specified by $G$ matched the true conditionals.

**Theorem A.4** *There exist a generator $G^*(\boldsymbol{y}, \boldsymbol{z})$ of which the conditional distributions $p_g(\boldsymbol{x}|\boldsymbol{y})$ and $p_g(\boldsymbol{x}|\boldsymbol{z})$ can both achieve equilibrium in their own minimax games $\mathcal{L}_{\boldsymbol{xy}}$ and $\mathcal{L}_{\boldsymbol{xz}}$.*

*Proof.* We can let $G^*(\boldsymbol{y}, \boldsymbol{z})$ characterize a valid distribution with their two integrals in Eq.1 equates $p_g^*(\boldsymbol{x}|\boldsymbol{y})$ and $p_g^*(\boldsymbol{x}|\boldsymbol{z})$. Under the nonparametric assumptions, the neural network estimator $G$ exists.

**Theorem A.5** *Mimimizing $\mathcal{R}_{\boldsymbol{z}}$ w.r.t. $I$ will keep the equilibrium of the adversarial game $\mathcal{L}_{\boldsymbol{xz}}$. Similarly, mimimizing $\mathcal{R}_{\boldsymbol{y}}$ w.r.t. $C$ will keep the equilibrium of the adversarial game $\mathcal{L}_{\boldsymbol{xy}}$ unchanged.*

*Proof.* Note $\mathcal{R}_{\boldsymbol{z}} = KL(p_g(\boldsymbol{x}, \boldsymbol{z})||p_i(\boldsymbol{x}, \boldsymbol{z})) - \mathbb{E}_{(\boldsymbol{x}, \boldsymbol{z}) \sim p_g(\boldsymbol{x}, \boldsymbol{z})}[\log \frac{p_g(\boldsymbol{z}, \boldsymbol{x})}{p(x)}]$, where the $KL$-divergence is always non-negative. As the second term is a constant to $I$, the optimum is obtained if and only if $p_g(\boldsymbol{x}, \boldsymbol{z}) = p_i(\boldsymbol{x}, \boldsymbol{z})$. The proof for $\mathcal{R}_{\boldsymbol{y}}$ is similar.

## B    Image Generation on MNIST and SVHN

We compare real samples with generated samples by SGAN on MNIST and SVHN. SGAN trained on MNIST uses 50 labels and SGAN trained on SVHN uses 1000 labels. The results are presented in Figure 1 and Figure 2.

---

[*] indicates equal contributions. [†] indicates the corresponding author. 31st Conference on Neural Information Processing Systems (NIPS 2017), Long Beach, CA, USA.

(a) Real      (b) Randomly generated      (c) Conditionally generated

Figure 1: Comparing real samples and generated samples on MNIST: (a) samples from MNIST test set, (b) randomly generated samples from SGAN, (c) conditionally generated samples from SGAN: each row has the same $y$ while each column shares the same $z$.

(a) Real      (b) Randomly generated      (c) Conditionally generated

Figure 2: Comparing real samples and generated samples on SVHN: (a) samples from SVHN test set, (b) randomly generated samples from SGAN, (c) conditionally generated samples from SGAN: each row has the same $y$ while each column shares the same $z$.

## C    Semi-supervised Generation on CIFAR-10 and Comparisons with ImprovedGAN

We compare samples generated from ImprovedGAN with feature matching, ImprovedGAN with minibatch discrimination and SGAN in Figure 3.

(a) ImprovedGAN (FM)      (b) ImprovedGAN (MD)      (c) SGAN

Figure 3: Visual comparison of generated images for (a) ImprovedGAN with feature matching, (b) ImprovedGAN with minibatch discrimination, (c) SGAN. For (c), each row shares the same $y$.

## D  Semi-supervised Class-conditional Generation on CIFAR-10

We perform conditional generation based on the semi-supervised trained SGAN models using 4000 labels of CIFAR-10 images. The results are shown in Figure 4.

(a) airplane      (b) automobile      (c) dog

(d) horse      (e) ship      (f) truck

Figure 4: Semi-supervised conditional generation on some classes of CIFAR-10.

## E  Semi-supervised Generation on CIFAR-10 with Advanced GAN Training Strategies

We generate some samples by SGAN trained with WGAN and the gradient penalties [1, 3], as listed in Figure 5.

## F  Network Architectures

We directly release our code on Github (`https://github.com/thudzj/StructuredGAN`) which contains the neural network architectures we used for MNIST, SVHN and CIFAR-10. Please refer to our code for the specific details.

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

(a) Random samples        (b) Random samples        (c) Random samples

(d) automobile          (e) bird          (f) cat

(g) dog          (h) horse          (i) ship

Figure 5: CIFAR-10 samples using training strategies.