[Reviews · NeurIPS 2017]

Reviewer 1



Summary: This paper proposes a novel GAN structure for semi-supervised learning, a setting in which there exist a small dataset with class labels along with a larger unlabeled dataset. The main idea of this paper is to disentangle the labels (y) from the hidden states (z) using two GAN problems that represent p(x,y) and p(x,z). The generator is shared between both GAN problems, but each problem is trained simultaneously using ALI[4]. There are two adversarial games defined for training the joints p(x, y) and p(x, z). Two "collaborative games" are also defined in order to better disentangle y from z and enforce structure on y. Pros: 1) Overall, the paper is written well. All the steps are motivated intuitively. The presented ideas look rational. 2) The presented results on the semi-supervised learning task (Table 2) are very impressive. The proposed method outperforms previous GAN and VAE works. The qualitative results in terms of generative samples look very interesting. Cons: 1) I found the mutual predictability measure very ad-hoc. There exist other measures such as mutual information that have been used in past (e.g. infoGAN) for measuring disentanglability of DGMs. 2) The collaborative games presented in Eq.4 and Eq.5 is equivalent to the mutual information term introduced in infoGAN[2] (with \lambda=1). In this case, the paper can be considered as the combination of infoGAN and ALI applied to semi-supervised learning. This slightly limits the technical novelty of the paper. Questions: 1) I am wondering why the adversarial game L_{xy} is only formed on the labeled set X_l. A similar game could be created on the unlabeled set. The only difference would be that instead of labels, one would use y=C(x) for the first term in Eq.3. However in this paper, instead, a collaborative game R_y is introduced. 2) Theorem 3.3 does not state whether minimizing R_z or R_y w.r.t G will change the equilibrium of the adversarial game or not. In other words, with minimization over G in both Eq.4 and Eq.5, can we still guarantee that G will match the data distribution. Minor comments: 1) line 110: Goodfellow et al is missing the reference. 2) line 197: Is l-2 distance resulted from assuming a normal prior on z? 3) line 213: p_g(z|y) --> p_g(x|y) 4) With all the different components (p, p_g, p_c, etc) it is a bit challenging to follow the structure of adversarial and collaborative games. A figure would help readers to follow the logic of the paper easier.

Reviewer 2



This paper proposes the SGAN model which can learn an inference network for GAN architectures. There are two sets of latent variables, y for the label information and z for all other variations in the image (style). The generator p_g conditions on y and z and generate an image x. The adversarial cost L_xz, uses an ALI-like framework to infer z, and the adversarial cost L_xy uses another discriminator to train a standard conditional GAN on the few supervised labeled data by concatenating the labels to both the input of the generator and the input of the discriminator. The SGAN network also uses R_y and R_z to auto-encode both y and z latent variables. R_y has an additional term that minimizes the cross-entropy cost of the inference network of y on the labeled data. To me, the core idea of this algorithm is very similar to the line of works such as Info-GAN [1] which learns a conditional GAN by auto-encoding the latent variable using an additional decoder network. The cost function R_z and R_y of SGAN are auto-encoding the latent variables in a similar fashion. Unlike what the paper mentions "However, the semantic of each disentangled dimension [in info-gan] is uncontrollable because it is discovered after training rather than designated by user modeling.", it is straightforward to incorporate label information in Info-GAN by additionally training the decoder to minimize the reconstruction over the supervised label pairs of (x,y). This is actually what the Improved-GAN paper [2] does and is similar to the first term of R_y in the SGAN model in equation 4. However, both of the Info-Gan and Improved-GAN models do not bother to learn the style variable z. The SGAN model, however, infers it by using an ALI-like technique to match p_i(x,z) to p_g(x,z) and thus making sure both the marginals match. In short, this paper does a good job in putting the architectures of the recent GAN works such as Info-GAN, Improved-GAN and ALI together to infer both the label and style, and use the resulting model in semi-supervised learning and style transfer. However, I didn't find the ideas of the paper significantly original. [1] Chen, Xi, et al. "Infogan: Interpretable representation learning by information maximizing generative adversarial nets." [2] Salimans, Tim, et al. "Improved techniques for training gans."

Reviewer 3



The paper presents a new flavor of generative adversarial nets that is more robust to perform conditional generation when limited labeled data is available. The paper is well written and I could follow the explanation of the method and experiments without issues. The experimental results are quite interesting both in terms of semi-supervised accuracy for image classification and in terms of “disentanglability” of y and z. Some questions/comments: - The mutual predictability (MP) metric proposed by the authors seems sound, although I don’t think that the comparison (in Fig. 1) with VAE trained without labeled information is fair. I think it would be better to compare with a semi-supervised VAE. - Are the images presented in Fig. 3 (b) and (c) randomly selected? Visually is hard to tell if SGAN is better than TripleGAN. Although the authors have shown that SGAN produces better inception score. - The authors should have included more details about the neural network architectures used. It could have been included as supplemental material. They mentioned that they “mostly follow those used in baseline works”, but which baseline work are they referring to? TripleGAN? Improved GAN? - SGAN seems to be quite expensive to train when compared to Improved GAN or even TripleGAN. The authors should have mentioned some statistics about the training time / convergence speed of the proposed method. Moreover, the authors should be more specific about the training process. For instance, how exactly are they increasing the proportion of synthetic samples throughout the training?